# CLiT: Combining Linking Techniques for Everyone

Kristian Noullet[1][0000−0002−4916−9443], Samuel Printz[2][0000−0003−1336−8536], and
Michael Färber[1][0000−0001−5458−8645]

Karlsruhe Institute of Technology
{kristian.noullet,michael.faerber}@kit.edu
samuel.printz@student.kit.edu

**Abstract.** While the path in the field of Entity Linking (EL) has been
long and brought forth a plethora of approaches over the years, many of
these are exceedingly difficult to execute for purposes of detailed analysis.
In many cases, implementations are available, but far from being a *plug-
and-play* experience. We present Combining Linking Techniques (CLiT),
a framework with the purpose of executing singular linking techniques
and complex combinations thereof, with a higher degree of reusability,
reproducibility and comparability of existing systems in mind. Further-
more, we introduce protocols for the exchange of sub-pipeline-level in-
formation with existing and novel systems for heightened out-of-the-box
compatibility. Among others, our framework may be used to consoli-
date multiple systems in combination with meta learning approaches
and increase support for backwards compatibility of existing benchmark
annotation systems.

**Keywords:** Entity Linking · Meta-Learning · Reproducibility · NLP ·
Semantic Web.

## 1 Introduction

The domain of Entity Linking (EL) deals with the interlinkage of textual men-
tions in text-based documents to corresponding entities in knowledge graphs.
Researching and developing EL systems is a highly time-consuming process, en-
compassing a multitude of considerations at each step, including a plethora of
moving parts – each capable of affecting the final results. Therefore, singling
out the reason for the success – or failure – through ablation studies oftentimes
constitutes a complex task, as any part of the processing pipeline may entail ma-
jor changes. Consequently, comparability to other systems is effectively rendered
*impossible* without tremendous research efforts. Even if such efforts are put in
for a single system, being able to make use of these for novel research may pose
an issue. In order to address these issues, we have worked on developing CLiT as
a highly modular and flexible framework, allowing for an ease of adoption into
existing systems and ones to come.
While research efforts allowing for performance evaluation of annotation tools

**Fig. 1.** Classical Pipeline for an EL system. Consisting of mention detection (MD), candidate generation (CG) and entity disambiguation (ED).

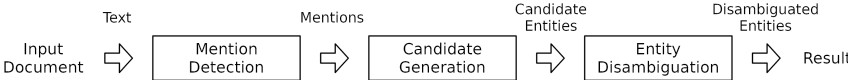

have been developed, easing the centralised execution of said systems for the purpose of further processing results has been mostly untouched. We intend to further extend the philosophy of increasing comparability between annotators through predefined evaluation data sets and computed metrics presented in [7, 5, 6], by enabling the use of complex workflows and in-depth analyses. In alignment with the vision of the Web of Data, all of our workflow's components and output provide and consume machine-readable data formats, in particular NIF 2.0 and JSON.

We intend to lead the research towards being able to answer the following research questions:

1. How can the research community **leverage** (sub-)component-level results from **existing systems**?
2. Can we increase result **explainability** for (mostly) black-box systems?
3. How may approaches be compared in an in-depth fashion? (**Comparability**)
4. How to properly reproduce existing systems? (**Reproducibility**)

To the best of our knowledge, no execution system attempting to fill the gaps of maximising reusability and comparability, additionally to minimising future development efforts for annotation approaches, exists. As such, we introduce CLiT, our means of simplifying life for and pleasing researchers as well as practitioners in the field of entity linking.

We advance the state-of-the-art by:

1. Introducing novel concepts for EL workflows, including compatibility with existing paradigms;
2. Allowing for nigh-infinite configurability of supported components in complex pipelines;
3. Enabling down-stream processing of annotation results rather than metrics;
4. Improving reusable components from existing systems and ones to come, increasing degree of system support;
5. Providing a knowledge graph agnostic and potentially multi-knowledge graph-supporting annotation service (through *translator*-subcomponents);
6. Defining open exchange protocols based on the Agnos [4] framework, JSON and NIF 2.0 for Mention Detection (MD), Candidate Generation (CG), Entity Disambiguation (ED) as well as *pre- and post-processing subcomponents* acting logically between the aforementioned;
7. Allowing simple introduction of existing systems through RESTful standards.

For further details on CLiT including a demonstration video, we refer interested parties to our Github page (https://github.com/kmdn/CLiTESWC2021).

## 2    System Design

*Classical Pipeline* While EL systems vary in terms of approaches and potential steps within respective pipelines, we identify the most commonly-employed ones as the *classical pipeline*. We use said pipeline as a template for our framework in order to reach compatibility with as many existing systems as possible. In Figure 1, we present our understanding of the functioning of a classical pipeline for a single system.

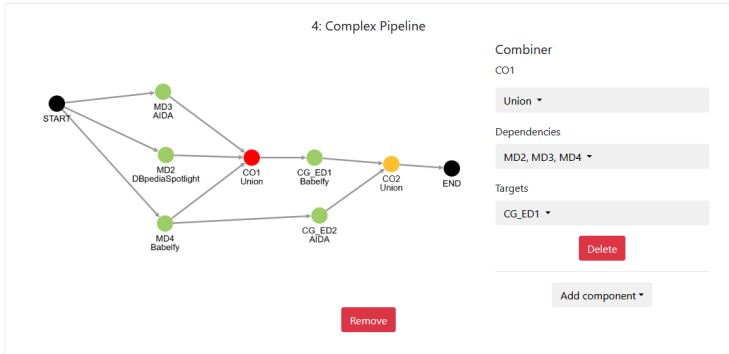

**Fig. 2.** GUI of CLiT with an example of a *complex* EL pipeline.

*Custom Processors* In order to allow for customized experiences and configurations, we introduce further processing capabilities with the intent of allowing for nigh-infinite combinations of system components (see Fig. 2). We refer to them as *processors* or *subcomponents*, handling post-processing of structures' output from prior tasks, preparing them for being, in turn, potentially further processed by subsequent steps in the chosen workflow. In this paper, we define 4 types of processors: *splitter*s, *combiner*s, *filter*s and *translator*s.

*Splitter* Allowing for processing of items prior to passing them on to a subsequent step, a splitter is utilised in the case of a *single* stream of data being sent to *multiple* components, potentially warranting specific splitting of data streams (e.g. people-related entities being handled by one system, while another processes movies). This step encompasses both a post-processing step for a prior component, as well as a pre-processing step for a following one. A potential post-processing step may be to filter information from a prior step, such as eliminating superfluous candidate entities or unwanted mentions.

*Combiner* As a counterpart to a splitter, a *combiner* subcomponent must be utilised in case multiple components were utilised in a prior step and are meant

to be consolidated through a variety of possible combination actions (e.g. union, intersection, ...). It combines results from multiple inputs into a single output, passing merged partial results on to a subsequent component.

*Filter* In order to allow removal of particular sets of items through user-defined rules or dynamic filtering, we introduce a subcomponent capable of processing results on binary classifiers: a *filter*. The truth values evaluated on passed partial results define which further outcomes may be detected by a subsequent component or translator.

*Translator* Enabling seamless use of annotation tools regardless of underlying Knowledge Graph (KG), the translator subcomponent is meant as a processing unit capable of *translating* entities and potentially other features, allowing further inter-system compatibility. It may be employed at any level and succeeded by any (sub-)component due to its ubiquitous characteristics and necessity when working with heterogeneous systems.

## 3    Conclusion & Future Work

In this paper, we introduced CLiT, a framework for the combination and execution of multiple entity linking approaches, both novel and existing. We show how components classically interact with each other based on a commonly-adopted pipeline and how they may be utilised, as well as extended through our framework. Currently, some annotators, such as Babelfy [1], DBpediaSpotlight [3] and AIDA [2] have been introduced to our framework – with more on the way. Furthermore, we will introduce semi-automated in-depth analysis features, allowing for collaborative evaluation, yielding a more fine-granular evaluation view on both annotators as well as data sets. Our contributions also increase the ease to train meta learning annotation classifiers with advanced degrees of flexibility and adaptability in relation to textual features.

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
