# OpenReview forum: "CLiT: Combining Linking Techniques for Everyone"
_eswc-conferences.org/ESWC/2021/Conference/Poster_and_Demo_Track — ESWC2021 P&D_

### Official Review · AnonReviewer3 · 2021-04-12
**Interesting implementation and conceptualization of a workflow manager lacking 1) comparison to other tools and 2) discussion of design decisions**

**Rating:** 5
**Confidence:** 4

**Review:**

The paper introduces a tool to combine and compare different components of entity linking workflows. The main idea behind this tool is to decompose existing EL workflows into individual components (types of such components are defined by authors) and then reuse the components to combine and benchmark different possible workflows. This work is still in progress, in future authors also promise to add "complex workflows and in-depth analyses" to provide additional insights into the linking results.

Overall the paper is **well written**. However, in the introduction authors  describe potential future plans and contributions. I struggled to make sense of these plans before I read the rest of the paper, therefore, I would suggest starting from the actual system itself. Also to better understand the main scientific contribution: "nigh-infinite configurability", "Improving reusable components", "simple introduction of existing systems through RESTful" it would be useful to have the description of the system first.

I think the work is **well motivated** -- though "comparability ... is ... impossible without tremendous research efforts" looks like too strong of a statement. The **main subject** of the paper is a software system with an openly available implementation, therefore **suitable for a demo session**. However, to **better estimate the scientific contribution** of the paper it would be **critically important** to overview the existing tools that pursue similar goals:

- GERBIL is a system that uses several included benchmark to compare performance of different annotator systems. https://aksw.org/Projects/GERBIL.html
- Silk is an actual linking framework that internally also defines workflows. From the point of view of workflow composition it could be seen comparable, though it does not focus particularly on EL task. http://silkframework.org/
- Teanga linked data based platform for natural language processing. Teanga also enables reuse of existing components. https://teanga.io/about
- FinTan allows to do data transformation for linguistic linked data and could be used to convert the outputs of different components to JSON-CONLL, NIF, etc. Fäth, Christian, et al. "Fintan-flexible, integrated transformation and annotation engineering." Proceedings of The 12th Language Resources and Evaluation Conference. 2020.

With respect to the **design of the system** itself it would be useful to have a holistic view on the decisions made by authors. The authors define 4 types of processors: splitters, combiners, filters and translators. It is not clear why authors choose exactly these components and if how many existing systems could be covered by this choice of components. Moreover, for splitters, that appear to me as potential first processor in a workflow, it is not clear how it could classify the data in order to enable "specific splitting of data streams (e.g. people-related entities being handled by one system, while another processes movies)".

**Anonymity:**

Yes, I would like my review to remain anonymous.

---

### Official Review · AnonReviewer2 · 2021-04-14
**CLiT: Combining Linking Techniques for Everyone**

**Rating:** 5
**Confidence:** 3

**Review:**

This demo paper presents a tool that allows to configure entity linking tools and run experiments with the given configuration.

The demo contains a graphical user interface that allows a user to dynamically connect the different tools.

It seems that tool follows a pipes & filters architecture style which is nice but the paper doesn't mention. I think it would make the paper more intuitive if the authors described the tool in that way.

The source code of the tool is available in github and it also contains 2 videos that demo the tool.

Given that the tool has a web frontend, I wonder why the authors didn't try to deploy it to some web server to make it easier to try the tool. Also, the authors didn't provide building instructions for the tool.

It is implemented in Java and uses Maven, so it probably follows a standard way to build it, but for non-java reviewers it would be better to provide some building instructions.

Minor detail: The paper contains definitions for MD, CD and EG, but not for CO (is it Combination?)

One feature that is not clear to me is how can a user configure the parameters of the different filters and components...is it possible?


**Anonymity:**

Yes, I would like my review to remain anonymous.

---

### Official Review · AnonReviewer4 · 2021-04-14
**Relevant, but added value of the implementation should be more emphasised**

**Rating:** 7
**Confidence:** 3

**Review:**

The approach proposes a framework for combining various entity linking tools in a pipeline in the context of a text annotation processing task. The authors describe the conceptual model of a system that allows defining complex workflows combining various plug-and-play modules.

Overall, the topic of the presented approach is relevant for the conference and, apparently, the paper describes an already implemented system. This should perhaps be better highlighted, as from the text it can appear that only a conceptual model is presented, while the GitHub link actually contains the implementation and instructions. Moreover, would be useful to have a use case example, where applying such a system is both justified (i.e., takes less effort than custom scripts) and sufficient (i.e., no custom scripts are required from the user).


**Anonymity:**

Yes, I would like my review to remain anonymous.

---

### Official Review · AnonReviewer1 · 2021-04-14
**Impressive Tool but Missing an Illustrative Use Case**

**Rating:** 6
**Confidence:** 3

**Review:**

The paper proposes CLiT, a framework to combine various entity linking techniques. It is based on workflows of annotation tasks for entity linking with defined pre- and post-processing steps.

While the software engineering of the tool is impressive, I missed a clear use case showing the versatility of the proposed solution. In addition, some aspects are not clear, e.g., what are meta-learning annotation techniques and how are they used in the proposed framework?

Other comments:
-page 3:the title of the subsubsections needs to end with a “.”, e.g., “Classical Pipeline.”, “Custom Producers”

**Anonymity:**

Yes, I would like my review to remain anonymous.

---

### Official Review · Program_Chairs · 2021-04-18
**Metareview: Accept (But requires clearer discussion of related tools)**

**Rating:** 6
**Confidence:** 5

**Review:**

The reviewers are generally positive about the impact that this work could have, but we do ask that the authors offer some (brief) discussion of the added benefits versus the tools mentioned by Reviewer 3.

**Anonymity:**

Yes, I would like my review to remain anonymous.

---

### Decision · Program_Chairs · 2021-04-19

Accept